# Self-Renewal of Macrophages: Tumor-Released Factors and Signaling Pathways

**DOI:** 10.3390/biomedicines10112709

**Published:** 2022-10-26

**Authors:** Serena Filiberti, Mariapia Russo, Silvia Lonardi, Mattia Bugatti, William Vermi, Cathy Tournier, Emanuele Giurisato

**Affiliations:** 1Department of Biotechnology Chemistry and Pharmacy, University of Siena, 53100 Siena, Italy; 2Department of Molecular and Translational Medicine, University of Brescia, 25100 Brescia, Italy; 3Department of Pathology and Immunology, Washington University School of Medicine, St. Louis, MO 63130, USA; 4Division of Cancer Sciences, School of Medical Sciences, Faculty of Biology, Medicine and Health, The University of Manchester, Manchester M13 9PL, UK

**Keywords:** tumor-associated macrophages, self-renewal, metabolic signature, signaling pathways, proliferation

## Abstract

Macrophages are the most abundant immune cells of the tumor microenvironment (TME) and have multiple important functions in cancer. During tumor growth, both tissue-resident macrophages and newly recruited monocyte-derived macrophages can give rise to tumor-associated macrophages (TAMs), which have been associated with poor prognosis in most cancers. Compelling evidence indicate that the high degree of plasticity of macrophages and their ability to self-renew majorly impact tumor progression and resistance to therapy. In addition, the microenvironmental factors largely affect the metabolism of macrophages and may have a major influence on TAMs proliferation and subsets functions. Thus, understanding the signaling pathways regulating TAMs self-renewal capacity may help to identify promising targets for the development of novel anticancer agents. In this review, we focus on the environmental factors that promote the capacity of macrophages to self-renew and the molecular mechanisms that govern TAMs proliferation. We also highlight the impact of tumor-derived factors on macrophages metabolism and how distinct metabolic pathways affect macrophage self-renewal.

## 1. Introduction

Macrophages comprise a heterogeneous and functionally versatile population of innate immune cells [1,2,3]. Every adult tissue contains an abundant population of macrophages either as resident cells or monocyte-derived cells that play tissue-specific functions and are essential for organ homeostasis [4,5,6]. Tissue-resident macrophages (TRMs) are long-lived cells and can be phenotypically distinct from monocyte-derived macrophages (MDMs). Initially, it was believed that TRMs in an adult are maintained by a constant replenishment of bone marrow-derived circulating monocytes in the steady state. This model was based on the assumption that terminally differentiated macrophages were unable to enter the cell cycle [7]. This was refuted by strong evidence that differentiated macrophages in several tissues were derived from precursors in yolk sac or fetal liver, independently of bone marrow, and self-maintained throughout life by local proliferation [8,9,10,11]. Accordingly, mature macrophages can proliferate in response to specific stimuli indefinitely and can also be expanded and maintained in long-term culture without loss of differentiation [12]. These findings demonstrated that macrophages exhibit a self-renewal potential similar to that of stem cells [12,13]. Distinct lineage-specific enhancer platforms regulate a shared network of genes that control self-renewal capacity in both stem and mature cells. In particular, single cell analyses identified key transcription factors, e.g., c-myelocytomatosis (c-Myc), Krüppel-like factor 2 (KLF2) and Krüppel-like factor 4 (KLF4), involved in macrophage self-renewal [14]. Conversely, MafB and c-Maf repress the self-renewal of resident macrophages [14,15]. Moreover, several in vivo studies have demonstrated that bone marrow-derived monocytes can also differentiate into self-renewing tissue-resident macrophages [16,17,18,19]. Similar to embryo-derived macrophages, KLF2 was strongly upregulated in the self-renewing bone marrow-derived macrophages. This was accompanied by the downregulation of MafB, unravelling a molecular mechanism of proliferation amongst self-renewing macrophages of different origins [13].

The discovery that TRMs were capable of self-renewal led to the new concept that macrophage proliferation played a key role in the expansion of macrophage populations in the TME, which is composed of the stroma and the tumor cells. During tumor growth, both TRMs and newly recruited monocyte-derived macrophages can give rise to TAMs which have been correlated with poor clinical outcome in most cancers [20,21,22,23,24,25,26]. Emerging evidence showed that TAMs retained the self-renewal capacity [23,25,27,28,29,30]. This unique feature was shown to markedly contribute to increasing the pool of protumoral TAMs, whether they derive from the yolk sac or the bone marrow, in mammary [23] and pancreatic [25] tumors.

High plasticity is a characteristic feature of macrophages that enables them to rapidly change phenotypes in response to environmental cues [31]. One of the best examples is the macrophage adaptation to the tumor environment, which has commonly been discussed in terms of a spectrum of polarization states from anti-tumor M1 to protumor M2 phenotypes [32]. Macrophage plasticity also concerns their metabolism and its modulation has emerged as a key factor in controlling macrophage activation and functions [33,34,35]. Although metabolic plasticity is associated with and participates to macrophage polarization state [36], it is becoming clear that specific metabolic signatures have an impact on self-renewing macrophages. In this review, we update the current knowledge on microenvironmental factors that regulate macrophages self-renewal ability and the molecular mechanisms underpinnings TAMs proliferation. We also discuss the importance of tumor-derived factors on macrophages metabolism and how distinct metabolic pathways are involved in macrophage self-renewal.

## 2. Macrophages Proliferation in Health and Disease

Macrophages are considered key players in innate and adaptive immune response, and in tissue repair. They protect the organism from infection both directly by pathogens phagocytosis and indirectly by acting as antigens presenting cells. Moreover, in light of recent discoveries, it has become obvious that resident macrophages derived from temporally and spatially distinct hematopoietic waves are critically important for tissue homeostasis [1,37]. Macrophages display different microanatomical localization in tissues and exhibits distinct rate of replacement via monocyte-derived and/or self-renewal capabilities [38]. The first wave of hematopoietic stem cells (HSCs) in human is detected at around day 18–19 of estimated gestational age (EGA) in the blood islands of the yolk sac (YS) where stem cells restricted to myelo-erythroid development are produced. These primitive macrophages are the first wave of colonization of the brain and other fetal organs, and microglial cells are the only macrophages that predominantly arise from yolk sac progenitors [39,40]. From 5 to 7 weeks of EGA, a second wave of hematopoietic cell development occurs in the aorta–gonads–mesonephros (AGM) when hematopoiesis temporally moved to the fetal liver. With the exception of microglia in the brain, most TRMs originate from monocytes produced in the fetal liver from around 4–5 weeks of EGA until 22 weeks of EGA [40]. Finally, at 10.5 weeks of EGA, hematopoiesis is definitively established in the bone marrow to give rise to monocyte-derived macrophages in adulthood [39,40,41]. To summarize, most adult tissue-resident macrophages are settled before birth and are maintained locally throughout adulthood by self-renewal. Adult bone marrow-derived circulating monocytes can be recruited to inflammatory sites to support TRMs populations [25].

Interestingly, the number of macrophages can increase as a consequence of proliferation, leading to chronic low-grade inflammation. This phenomenon can cause a variety of alterations associated with certain pathological conditions [42,43]. For example, under normal conditions, adipose tissue macrophages (ATMs) whose function is to remove cellular pathogens, dispose of dying adipocytes and process lipids, constitute approximately 10% of all cells in fat tissue. In obesity, the proportion of ATMs increases to 50%, thereby leading to a persistent inflammatory state that results in adipose tissue fibrosis, insulin resistance and type-2 diabetes mellitus (T2DM) caused by β-cell dysfunction in pancreatic islets [43]. Initially, increased ATM numbers were believed to be a result of blood monocyte recruitment in the affected tissues. This view has changed in light of recent evidence that the accumulation of ATMs in crown-like structures around dead adipocytes is mainly a consequence of in situ proliferation at the early stage of obesity and is further promoted by the recruitment of monocytes at the late stage [43]. Pancreatic inflammation associated with obesity is characterized by the accumulation of immune cells and elevated production of inflammatory cytokines and chemokines [44,45,46]. A high number of islet-resident macrophages is a typical hallmark of T2DM associated with obesity [42,47]. Interestingly, intra-islet macrophages proliferate and expand locally, independent of recruitment from circulating monocytes, and may impair β cell function by restricting insulin secretion, suggesting a complex crosstalk network between proliferating macrophages and β cells.

Macrophages with self-renewal capabilities have also been involved in non-resolving inflammation associated with cancer. TAMs represent the most abundant inflammatory cells in the TME [48,49,50]. Crosstalks between macrophages and the other cells of the TME, via cell-to-cell contact or the production of soluble factors, are essential for promoting cancer cell proliferation, angiogenesis, and metastasis [3,51,52]. The accumulation of TAMs in tumors results from the constant recruitment of circulating monocytic precursors from the blood [23,53]. However, recent studies revealed that pools of TAMs can originate from proliferating TRMs [23,28,29,30,54]. This is significant given that proliferative macrophages were detected in sarcomas [28], fibrosarcoma [55], gastric cancer [56], colorectal carcinoma CRC [57], prostate cancer [58], non-small cell lung cancer (NSCLC) [59,60], ovarian cancer [61], CNS cancer [62], breast carcinomas [29,30], and pancreatic cancer [25]. In a murine model of human pancreatic ductal adenocarcinomas (PDAC), circulating monocytes were found to be dispensable for tumor growth, while the expansion of TRMs through in situ proliferation played a critical role to regulate TAMs population supporting cancer progression [25]. In addition, whereas monocyte-derived TAMs played a more potent roles in antigen presentation, proliferating embryonically derived macrophages exhibited a profibrotic transcriptional profile [25], indicative of their role in producing and remodeling molecules in the extracellular matrix. More recently, using tissue microarrays in a large set of human primary malignant lesions, we demonstrated a noticeable increase in the number of proliferating macrophages (Ki67^+^CD163^+^ cells) compared with normal tissues [63] (Figure 1). Moreover, TAMs accumulation through proliferation was a significant feature of malignancy, clearly indicating that local macrophage proliferation was a common hallmark of human tumors and a potentially important prognostic marker of malignancy [63].

## 3. Tissue Microenvironment and Self-Renewal of Macrophages

The proportion of macrophages derived from bone marrow-derived circulating monocytes or TRM varies according to the organ and to the presence of any pathology [64]. Macrophages are tightly controlled by the local tissue microenvironment in response to a variety of soluble factors, including colony-stimulating factor 1 (CSF-1), which acts through the CSF-1 receptor (CSF-1R) expressed at the cell surface of all macrophages [12]. Macrophage depletion in animals lacking CSF1 (the osteopetrotic op/op mutant mice or toothless tl/tl mutant rats) provided the first genetic evidence that CSF-1 was essential for determining macrophage number [40,65]. The trophic function of CSF-1 was confirmed through the demonstration that antagonist CSF-1 antibodies blocked the proliferation of resident macrophages in homeostasis and inflammation in the peritoneal cavity, spleen, lung, and in the growing myometrium during pregnancy [12]. More recently, mature alveolar macrophages (AM) were shown to self-renew throughout the lifespan in response to CSF-1 produced by resident lung epithelia and fibroblasts [66].

Another important regulator of macrophage number in tissues is granulocyte/macrophage colony-stimulating factor (GM-CSF), also known as colony-stimulating factor 2 (CSF-2). In particular, the depletion of GM-CSF impaired the self-renewal of tissue macrophages derived from the fetal liver [67]. GM-CSF was also shown to be involved in the proliferation of peritoneal macrophages in vivo and in maintaining the homeostasis of AM [68]. Moreover, the administration of IL-4 enhanced macrophage proliferation in the liver, spleen, and bone marrow [8]. Similarly, the administration of IL-4 increased the in situ proliferation of ATMs in lean mice, suggesting that IL-4 could play role in driving the local proliferation of ATMs in obesity [43]. Accordingly, one study showed that IL-6 acted as a Th2 cytokine in obesity through upregulating the IL-4 receptor α [69]. Moreover, monocyte chemotactic protein 1 (MCP-1) and Osteopontin (OPN) have been shown to drive monocyte chemotaxis and stimulate macrophage proliferation in adipose tissue [70,71]. More recently, one study showed that CCL2-mediated activation of CCR2 regulated monocytes and macrophages proliferation in skin wounds, under diabetic conditions [72]. These proliferative macrophages exhibited upregulation of genes associated with progression and regulation of cell cycle [72].

Additionally, IL-34 produced by neurons and keratinocytes can interact with the CSF-1R to promote the self-renewal of microglia and Langerhans cells in the brain and in the skin under homeostatic conditions [12,40,66]. Recently, the involvement of another cytokine, IL-24, was involved in macrophage proliferation. This study showed that IL-24 receptors are highly expressed on decidual macrophages and M2-like macrophages in early pregnancy, and it plays a central role in cell growth and steady-state regulation for these ones, demonstrating that decidual macrophages are capable of self-renewal education by IL-24, which is released from decidual stroma cells [73].

### Macrophages Proliferation in the Tumor Microenvironment (TME)

It is now established that TAMs are able to proliferate in the TME, in response to CSF-1 [12,74,75]. (Figure 2). Multiple studies have demonstrated that CSF-1/CSF-1R is highly expressed in several tumor tissues such as breast cancer [76,77,78], prostate cancer [79], head and neck cancer [80,81], ovarian cancer [82], gastrointestinal cancer [83], colon cancer [84,85], pancreatic cancer [86], and mesothelioma [87]. Previous studies conducted in mouse model and in patients demonstrated that antibody blockade of CSF-1R dramatically decreased the number of TAMs in tumor tissues [88,89]. The dramatic reduction in TAMs density was a consequence of reduced monocyte recruitment, but also decreased macrophage proliferation, as demonstrated by reduced Ki67 staining. A further study showed that CSF-1 blocking significantly reduced the percentage of proliferating TAMs, suggesting that stimulation of CSF1R played a major role in TAMs proliferation [90]. Thus, the CSF-1/CSF-1R signaling axis has become an attractive target to decrease the number of TAMs in tumors [91]. However, although the proliferation of macrophages can be influenced by a high expression of CSF-1, other factors can be involved in mediating CSF-1R signaling [64]. For example, IL-34 expression was detected in giant cell tumors of bone [92], in primary lung cancer tissues [93], in hematological malignancies, brain, breast, neck, biliary, and ovarian cancer [94,95]. High IL-34 expression has been documented in various cancers where it plays important roles in multiple aspects of the tumorigenesis [92,96,97,98]. IL-34 stimulates CSF1R in TAMs to promote their survival and in monocytes to promote their recruitment to the tumor area [50]. Accordingly, neutralization of CSF-1 together with or without IL-34 reduced the total number of immune cells within tumors, and this reduction led to a significant reduction in TAMs. More specifically, the treatment with anti-CSF-1 significantly influenced TAM proliferation, while IL-34 neutralization alone partially reduced the percentage of Ki67^+^ TAMs, suggesting that IL-34 plays a key role in the macrophages recruitment and function but that only partially affects TAMs proliferation [90]. Besides to regulate TAMs, accumulating evidence suggest that IL-34 stimulates cancer cell proliferation through its ability to interact with CSF1R in cancer cells [99].

In the last few years, several additional soluble factors have been discovered capable of inducing macrophage self-renewal in the TME (Table 1). One of these important immunomodulatory factors is tumor-derived purine nucleoside adenosine (ADO) [100]. ADO is synthesized from ATP, ADP, AMP, or nicotinamide derivatives by ectonucleotidases (CD39 and CD73) and released in the extracellular space through nucleoside transporters [101]. The accumulation of ADO is further regulated through the conversion of adenosine into inosine by adenosine deaminase enzymes (ADA) [102]. Recently, it has been discovered the secretion of ADA2 by monocytes enhances their differentiation into macrophages and the proliferation of monocytes and macrophages via autocrine signaling [103,104] (Figure 2). In addition, ADA2 promoted the polarization of macrophages into M2-like phenotypes in triple-negative breast cancer (TNBC) [105]. At the molecular level, ADO acts by binding G protein-coupled adenosine receptors (ARs) A1, A2a, A2b, and A3. The stimulation of A2 receptors expressed at the cell surface of macrophages decreased TNFα expression while inducing the secretion of IL-10 [101,105]. Evidence that the level of expression of A2A receptors in macrophages increased after exposure to tumor culture supernatants from hepatoma suggested that ADO promotes macrophage proliferation, in part, through the transcriptional regulation of A2A [106]. Additionally, ADO functioned synergistically with GM-CSF to stimulate macrophage proliferation in hepatocellular carcinoma (HCC) and regulated human macrophage self-renewal via PI3K/Akt and MEK/ERK signaling downstream of A2A activation [106].

Another mechanism that could control the proliferation of macrophages involves the triggering receptor expressed on myeloid cells 2 (TREM2). TREM2 is a single-pass transmembrane receptor of the immunoglobulin superfamily that binds phospholipids as well as anionic molecules, including bacterial products, DNA, and lipoproteins [112,113].

TREM2 transmits intracellular signals through the recruitment of the protein tyrosine kinase Syk via the adaptor proteins DNAX-activation protein 12 (DAP12; also known as TYRO for protein tyrosine kinase-binding protein) and DAP10 (also known as hematopoietic cell signal transducer) [110,112]. Several studies reported that the soluble form of TREM2, i.e., sTREM2, promoted microglial survival through the activation of PI3K/Akt, ERK1/ERK2, and p38 [112,114]. Although most of the information about TREM2 are from microglia in central nervous system [115], TREM2 is expressed in TAMs and seems to play an immunosuppressive role in cancer [55,110]. An increase in TREM2^+^ macrophages has been observed in HCC [113] and in pancreatic ductal adenocarcinoma [116]. Martina M. et al. [110] also demonstrated that *Trem2^–/–^* mice exhibited resistance to tumor growth due to an alteration in macrophage subsets and an increase of intra-tumoral CD8^+^ T cells. However, the mechanism by which TREM2 deficiency impacted the fate of tumor macrophages remains unclear. Recently, it has been shown that TREM2^+^ TAMs preferentially expressed macrophage proliferation-related genes, as demonstrated by the increase in Ki-67 expression in this population [111] (Figure 2). In parallel, in vitro studies have revealed that bone marrow-derived macrophages generated from mice deficient in either TREM2 or DAP12 proliferated poorly [117,118] and that TREM2 expression was induced in human monocytes and mouse bone marrow cells exposed to CSF-1 [119]. Accordingly, CSF1 deprivation induced apoptosis in TREM2-deficient bone marrow-derived macrophages in vitro [120]. This effect could be reversed by the addition of soluble TREM2 (sTREM2), thereby suggesting that sTREM2 augmented survival of macrophages [112]. However, whether sTREM2 is involved in macrophages proliferation remains to be investigated.

## 4. Self-Renewal of Macrophages: Molecular Mechanisms

A complex transcriptional network is implicated in regulating the self-renewal of macrophages in a tissue-specific manner [107,108]. ERK1/2 is required for mediating the proliferation of bone marrow-derived macrophages in response to CSF-1 [121,122,123]. The activation of ERK1/2 downstream of CSF-1R and GM-CSFR activates transcription factors of the Ets family, which subsequently increase the expression of cyclin-D, directly or via c-Myc (Figure 3). Recently, ERK5-MAPK has been implicated in tumor macrophage proliferation through the genetic demonstration that myeloid ERK5 deficiency suppressed the proliferation of both resident and infiltrated macrophages in metastatic lung nodules [63]. Mechanistically, ERK5 maintained the capacity of TAMs to proliferate by suppressing p21 expression to halt their differentiation program, via a mechanism that might involve a suppression of c-Maf/MafB and increased expression of Ets2 and Cyclin-D [63]. Accordingly, proliferative TAMs in murine pancreatic ductal adenocarcinoma (PDAC) exhibited reduced levels of MafB and c-Maf, together with increased c-Jun and Ets2 expression [25].

Other transcription factors strongly upregulated in response to CSF-1 stimulation are KLF2 and KLF4. KLF2 and c-Myc appear to be the main drivers of CSF-1-induced bone marrow macrophage self-renewal [13]. In contrast, KLF4 does not induce macrophage proliferation but appears to mitigate the transforming activity of c-Myc by opposing its effect on p21 and p53 [12,14]. The JAK2/STAT5 axis constitutes another important pathway that mediates macrophage self-renewal in response to GM-CSF [12,14] and Gata6 is a master regulator of peritoneal macrophage proliferation in response to CSF-1 stimulation [124]. More recently, SIRT1 was identified as an important factor regulating the self-renewal gene network overlapping between macrophages and embryonic stem (ES) cells. A high level of SIRT1 expression during bone marrow-derived macrophage differentiation increased their proliferative capacity [125]. Moreover, the pharmacological inhibition of SIRT1 in vivo reduced steady-state and cytokine-induced proliferation of alveolar and peritoneal macrophages by inhibiting E2F1 and c-Myc [125].

Two new closely related transcription factors, namely Bhlhe40 and Bhlhe41, have been recognized as novel regulators of tissue-resident macrophages [107,108]. Specifically, the loss of Bhlhe40 in large peritoneal macrophages (LPMs) increased the level of c-Maf and MafB mRNA and lowered the expression of cell cycle-related transcripts [108]. In parallel, another study demonstrated that Bhlhe40/Bhlhe41-deficient AM were unable to self-renew in response to GM-CSF stimulation [107]. As previously described, Maf-B and c-Maf are master suppressors of many genes in the self-renewal gene network shared by both macrophages and ES cells [12,14,15]. Thus, Bhlhe40 repressed the expression of the transcription factors c-Maf and Mafb, thereby directly promoting expression of transcripts encoding cell cycle-related proteins to enable macrophages proliferation [108].

Since AM are derived from embryonic precursors and are self-maintained with minimal contribution from circulating bone marrow-derived precursors in steady states, much attention has been paid to understanding the molecular mechanisms underlying their self-maintenance. Lkb1-deficiency impaired the ability of AM to self-renew [127]. Another important role in AM homeostasis is played by mTORC1 signaling, which exerts its function at least in part by ensuring optimal responsiveness of AM to GM-CSF-induced cell cycle entry and regulates the repopulation of AM post-irradiation-induced replenishment [128]. Very recently, Daniel Bakopoulos and collaborators [129] provided evidence that PDGF- and VEGF-receptor-related pathway regulate macrophages self-renewal in Drosophila, supporting a novel mechanism for the regulation of macrophage proliferation by the dynamic transcriptional control of PDGF- and VEGF-related factor 2. The relevance of this pathway in mammals’ macrophages proliferation is still unknown.

Macrophage self-renewal appears even more complex in pathological conditions. IL-4/STAT6 signaling is the major driving force for ATM proliferation, as demonstrated by genetic evidence that STAT6-deficient ATMs are unable to proliferate in response to IL-4 [43]. However, how self-renewal is controlled by endocrine signals during obesity is still largely unexplored. A recent report showed that bisphenol A (BPA; 2,2-bis(4-hydroxyphenyl-propane), which accumulates in the adipose tissue and contributes to obesity-associated diseases, increased macrophage self-renewal. The molecular mechanism of BPA-induced macrophage self-renewal is associated with the activation of ERK1/2 and an increase in the levels of liver X receptor alpha (LXRα), but further studies will be needed to evaluate whether LXR expression serves as an indicator for in situ proliferation of tissue-resident macrophage [130].

## 5. Metabolism Signature Associated to Self-Renewal of Macrophages

Recent studies investigating macrophage metabolic reprogramming have shown that macrophages are capable of tightly coordinating a switch of their metabolic programs, according to their immediate energy and proliferative state [131,132]. M1/M2 classification has allowed the demonstration of how metabolic plasticity is associated with and participates in macrophage polarization [36]. From a metabolic point of view, it has been shown that proinflammatory macrophages (M1) have peculiar metabolic demands to produce nitric oxide (NO), reactive oxygen species (ROS), and proinflammatory cytokines [133,134]. Thus, to meet their energy requirements M1 macrophages utilize largely glycolysis, the pentose phosphate pathway (PPP), and fatty acid synthesis (FAS), whereas tricarboxylic acid (TCA) cycle is truncated and mitochondrial oxidative phosphorylation (OXPHOS) downregulated [131,135,136,137]. Lipopolysaccharide (LPS) stimulation causes the accumulation of citrate and succinate metabolites, leading to HIF1α stabilization, a major driver of glycolytic gene expression [131], which, in turn, sustained IL-1β and ROS to enhance the M1 macrophage inflammatory response [135,138,139]. The latter is also determined by arginine metabolism as M1 macrophages upregulate nitric oxide synthase (iNOS), which metabolizes L-arginine into L-citrulline and NO [133,140]. Additionally, it has been reported that macrophages exit from the cell cycle during M1 differentiation, while they sustain high metabolic activity imposed by the production of bactericidal factors, suggesting a potential coordination between metabolic regulation and macrophage physiology [141,142,143].

In contrast, anti-inflammatory M2 macrophages are more dependent on OXPHOS activity, and their metabolic profile supports their cell proliferation and promotes tissue repair [8,139,144,145]. As a matter of fact, M2 macrophages use fatty acid oxidation (FAO) and OXPHOS pathways to meet their ATP requirements [34,146,147,148]. In response to IL-4 stimulation, JAK-STAT6 signaling is activated, and the transcription factor PPARγ-coactivator-1β (PGC-1β) is expressed, inducing M2 macrophage reprogramming towards FAO and mitochondrial biogenesis [146,149,150]. Additionally, L-arginine metabolism is different in M2-like subpopulation; it is metabolized by the enzyme arginase 1 (Arg1) into ornithine and urea, and then, ornithine generates polyamines and proline, which have functional importance in cell proliferation and collagen synthesis [133,140,151]. However, accumulating evidence suggests that macrophage metabolism is not as simple as previously thought, and recent studies have shown that enhanced glycolysis, a key metabolic pathway of the M1 program, is also required for M2 activation. As demonstrated by Covarrubias et al., after IL-4 treatment, BMDMs increase both glycolysis and oxidative metabolism in an Akt-dependent manner, contributing to M2 gene induction. Using gene enrichment analysis of Akt- and ATP Citrate Lyase (ACLY)-coregulated genes, they also observed that cell cycle and DNA replication pathways are upregulated in M2 macrophages, indicating that IL-4 may stimulate macrophage proliferation in an Akt- and ACLY-dependent manner [109]. Similar results have been obtained by Stanley Ching-Cheng Huang et al., [35] who have shown that enhanced mTORC2 signaling, upstream of IRF4 expression, is critical to enhance glycolysis and mitochondrial pyruvate import, which are essential for M2 activation. In the same study, they also showed that pharmacological glycolytic inhibition (2-DG) suppressed the M2 peritoneal macrophages proliferation, suggesting that enhanced glucose metabolism is essential for M2 macrophage activation [35]. On the other hand, recent findings have shown that 2-DG may have additional off-target effects and that glycolysis is not mandatory for M2 activation if OXPHOS is intact, but it becomes necessary if OXPHOS is compromised [152]. Overall, these findings indicate that, while the M1-associated metabolism suppresses cell cycle progression, M2-activated metabolic pathways sustain cell proliferation, suggesting a coordination between metabolic regulation and macrophage self-renewal.

Macrophages are highly specialized in sensing the microenvironment and modify their properties accordingly, raising the question: How the environmental factors impact macrophage metabolism and self-renewal? Emerging evidence has shown that soluble factors and extracellular signaling events determine cell proliferation, in part by modulating metabolic programs [153,154]. The main protagonist of macrophages proliferation is CSF-1 and it has been reported that mitogenic stimulation of BMDMs with CSF-1 induces a Myc-dependent transcriptional program, which promotes cell proliferation and upregulation of glucose and glutamine catabolism [145]. Conversely, LPS stimulation of BMDMs, inhibits Myc expression and macrophages proliferation due to upregulation of HIF1α and increased expression of components of the glycolytic pathway. These data indicate a coordinated regulation of macrophage metabolic programs to support their growth and proliferation [145]. Accordingly, Ting Wang et al. [131] reported increased glycolysis and decreased mitochondrial oxidation in peritoneal macrophages isolated from mice over-expressing HIF1α in macrophages, thereby demonstrating that HIF1α-induced glycolysis is essential to the activation of inflammatory macrophages. Nevertheless, a recent study showed that c-Myc expression had not only a role in macrophages-induced proliferation by glucose metabolism but was also required in the early phase of inflammatory stimulation before the stabilization of HIF1α [155].

Macrophage subsets exist within different organ systems, suggesting that distinct environments shape macrophage metabolism and effector function. One example is the case of tissue-resident AMs, reside in the alveolar lumen, which originate from fetal yolk tissue, are a long-lived and self-renewing cell population. Recently, Zhu B. et al. [156] using a HIF-1α activity reporter mouse strain, demonstrated that, following viral infection, AMs activate a ‘‘nonconventional’’ Wnt-β-catenin-HIF-1α complex that simultaneously promotes glycolysis-dependent inflammation and suppresses AMs proliferation and self-renewal capabilities. Thus, these results could reflect the correlation between macrophage proliferation ability and OXPHOS pathway. Further important evidence supporting the correlation between macrophage oxidative phosphorylation and proliferation is given by Marie Pereira et al. [157], who analyzed the macrophage metabolic reprogramming due to iron deprivation. Using RNA-seq analysis, they showed that iron deficiency in human macrophages caused a downregulation of cell cycle-mitosis and OXPHOS pathways, plus a concomitant activation of HIF-1-glycolysis and IFN transcriptional responses. One of the pathways that is required for AMs proliferation is GM-CSF that plays a key role in multiple metabolic pathways mainly through the regulation of mitochondrial functions [158]. Wessendarp M. et al, using GM-CSF receptor-β-chain deficient (Csf2rb^−/−^) mice, demonstrated that compared to wild-type macrophages, Csf2rb^−/−^ macrophages had reduced levels of ATP, as well as glutathione and cellular ROS levels, and a greater level of cells with compromised cell cycle. These results indicate that GM-CSF stimulation is required for the maintenance of mitochondrial mass and function in macrophages, as well as for AM self-renewal.

TME represent one of the best examples of environment that shape macrophage metabolism and self-renewal. In a recent report, using metabolic autofluorescence imaging, Heaster TM et al. [159] quantified metabolic heterogeneity between macrophages within normal and cancerous mouse pancreatic tissues *in vivo*. Interestingly, they have shown that tumor-infiltrating macrophages prefer oxidative metabolism due to their decreased optical redox ratio (NAD(P)H divided by flavin adenine dinucleotide (FAD) intensity) compared to dermal macrophages. Additionally, using in vitro experiments, Boyer S. et al. [160] have performed multiomic characterization (i.e., transcriptomics, proteomics, metabolomics) of tumor-educated macrophages polarized with pancreatic cancer-conditioned media, to molecularly define pancreatic TAMs. Their proteomics and transcriptomics analysis have shown that TAMs contain the greatest proportion of upregulated metabolic enzymes compared to M0, M1, and M2. Among these enzymes Thioredoxin-interacting protein (TXNIP), ATP Citrate Lyase (ACLY), and Arg1 have been identified as unique contributors to TAMs metabolism [160].

In support of the link between metabolic profile and self-renewal capacity in TAMs, Zhang and Liu [111] demonstrated that a strong enrichment of fatty acid metabolism and protumor pathways were observed in TREM2^+^ TAMs population, which preferentially expressed macrophage proliferation-related genes, such as Ki-67, whereas immune-responsive and DNA repair pathways were significantly downregulated.

How the tumor environment impact macrophage metabolism and self-renewal? TME is a typically nutrient-poor microenvironment with high concentrations of lactic acid, derived from glycolysis and glutaminosis, accompanied by reduced pH and low glucose concentration [161,162]. Thus, recent studies reported that this altered metabolic profile of TME in combination with metabolic plasticity of macrophages and their intimate crosstalk with tumor cells directly drives macrophages to adopt immunosuppressive M2-like phenotype [163,164]. On the basis of recent publications, TAMs, similar to cancer cells, which upregulate aerobic glycolysis and lactate production (so-called Warburg effect), respond to the altered metabolic profile of TME by polarizing to a cellular state which utilizes glycolysis, FAS, FAO and altered glutamate metabolism [62,164]. For example, lactic acid induces VEGF and Arg1 expression in TAMs, thus promoting a pro-angiogenic signature, and potentiates glycolysis by activating the Akt/mTOR pathway. Once cancer grows, hypoxia induces in TAMs upregulation of REDD1 (regulated in development and in DNA damage response 1), an mTORC1 inhibitor, which subsequently inhibits the glycolytic shift toward oxidative metabolism. Thus, increasing glucose availability in the perivascular space causes endothelial hyperactivation, leading to neo-angiogenesis and metastasis [2,165,166] (Figure 4).

In a recent report, Irene Soncin et al. [168] showed that adenomas microenvironment alters the metabolic signature of TAMs as well as their proliferative capacity. Thus, using fate mapping studies of Ccr2^−/−^ mice but also RNA-seq and Ki-67 analyses, they showed that both F4/80^hi^MHCII^hi^ and MHCII^low^ macrophages expanded in situ in adenomas by self-renewal, independently from CCR2. In addition, their RNA-seq analysis demonstrated that F4/80^hi^MHCII^low^ macrophages express the highest levels of key glycolytic genes and Arg1 transcripts, typical for M2-polarized and protumor macrophages. These data reveal that tumor-resident F4/80^hi^MHCII^low^ upregulates several metabolic pathways and at the same time increased their proliferative capacity during tumor progression.

## 6. Conclusions

Macrophages are mature differentiated cells that may have a self-renewal potential similar to that of stem cells. In many mammalian tissues, major macrophage populations were found to be derived from embryonic progenitors and to renew independently of hematopoietic progenitors [12,66]. As in other chronic-inflammatory disease, macrophages can proliferate in cancer tissues. A large body of scientific evidence has implicated TAMs in tumor progression, metastasis, evasion of the immune response, and unfavorable response to therapy. TAMs respond to environmental signals by acquiring a wide spectrum of phenotypic and functional states. Several evidence indicates that, during tumor progression, TAMs predominantly show an M2-like phenotype [169] and maintain a self-renewal ability [63,170,171,172], which is correlated with tumor stage malignancy [63]. The glucose, amino acid, lipid, and iron metabolic profiles of macrophages have all been shown to be altered throughout tumor progression. The profiles of these metabolic pathways of macrophages are regulated by broad intercellular and intracellular signaling pathways involving several transcription factors. As previously mentioned, M2 macrophage metabolism depends on FAS and OXPHOS pathways, which are induced by mitogenic stimuli, such as CSF-1, which, in turn, induces their proliferation [145,166,167]. In TME, CSF-1 is one of the main growth factors released by tumor cells and stroma cells able to modulate macrophage physiology [173,174,175]. However, since macrophage proliferation in response to tumor supernatant might be not completely dependent on CSF-1R signaling [64], others tumor-released factors could be involved in TAMs self-renewal and activate specific metabolic signatures. Thus, in keeping with these findings, it would have to be investigated how metabolic reprogramming of TAMs could affect their self-renewal capacity. During the past decade, the explosive growth in macrophage-targeted therapy indicates that the TAM-targeted therapy is an effective antitumor strategy, especially as a complementary strategy in combination with conventional chemotherapy, radiotherapy, or immunotherapy. Most of the preclinical studies and clinical trials have been based on the therapeutic strategies according to their different mechanisms, including those that inhibit mononuclear macrophage recruitment, those that deplete TAMs, and those that reprogram TAMs [176,177]. However, clinical trials specifically designed to modulate or interfere with the self-renewal of TAMs are lacking. Understanding all possible involvements of signaling pathways and the metabolic modulation of TAMs could be an important strategy for future therapies, for example, based on macrophages reprogramming modulating at the same time their metabolism and self-renewal.

## Figures and Tables

**Figure 1 biomedicines-10-02709-f001:**
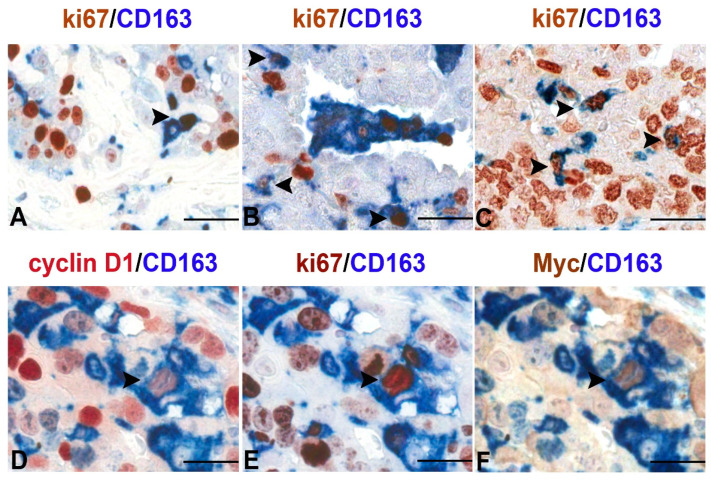
Proliferating TAMs in human cancers. Sections are from human pleural mesothelioma (**A**), lung adenosquamous carcinoma (**B**), serous ovarian carcinoma (**C**) and transitional bladder carcinoma (**D**–**F**) and stained as labelled. A variable fraction of CD163^+^ TAMs co-express the proliferation marker ki-67 as demonstrated by double immunostaining (**A**–**C**,**E**); a minor fraction of TAMs also co-express nuclear Myc and Cyclin D1 as revealed by sequential double immunostains (**D**–**F**). Images are obtained from 40× digitalized slides and resized using Adobe Photoshop. Scale bar (**A**–**C**), 44 μM; (**D**–**F**), 33 μM.

**Figure 2 biomedicines-10-02709-f002:**
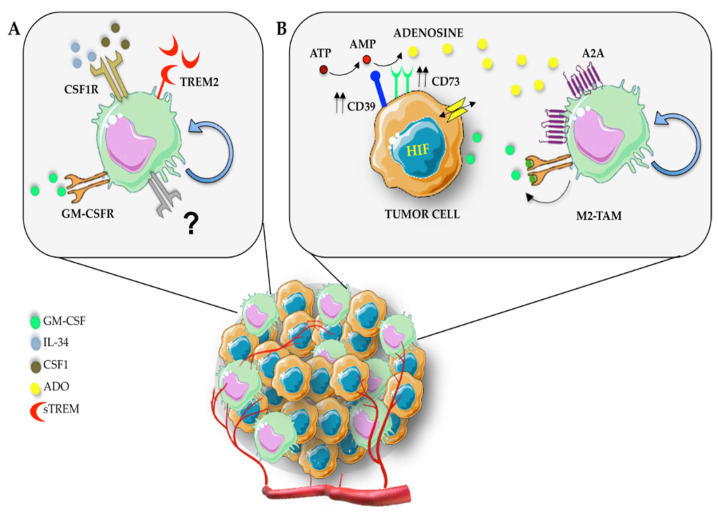
TAMs proliferation is regulated by tumor–environment soluble factors. (**A**) CSF-1 and, to a lesser extent, IL-34 are the main factors produced by tumor and stroma cells to induce TAMs proliferation through stimulation of the CSF-1R in several tumor types [63,90,93,95,99]. Additionally, GM-CSF induces macrophage proliferation in hepatocellular carcinoma (HCC) [106]. Although the importance of IL-4-driven macrophage self-renewal has been reported in inflammatory contexts [43], the direct role of IL-4 on TAMs proliferation remains to be clarified [64]. Recently, TREM2 was identified as a specific TAM marker [110] and increased proliferation-related genes (Ki-67) were observed in TREM2^+^ TAMs population from NSCLC samples [111]. (**B**). Tumor-derived purine nucleoside adenosine (ADO) accumulation [101] induces macrophage proliferation, working synergically with GM-CSF, and the increased A2A receptor expression on macrophage surfaces is crucial to support this process [106].

**Figure 3 biomedicines-10-02709-f003:**
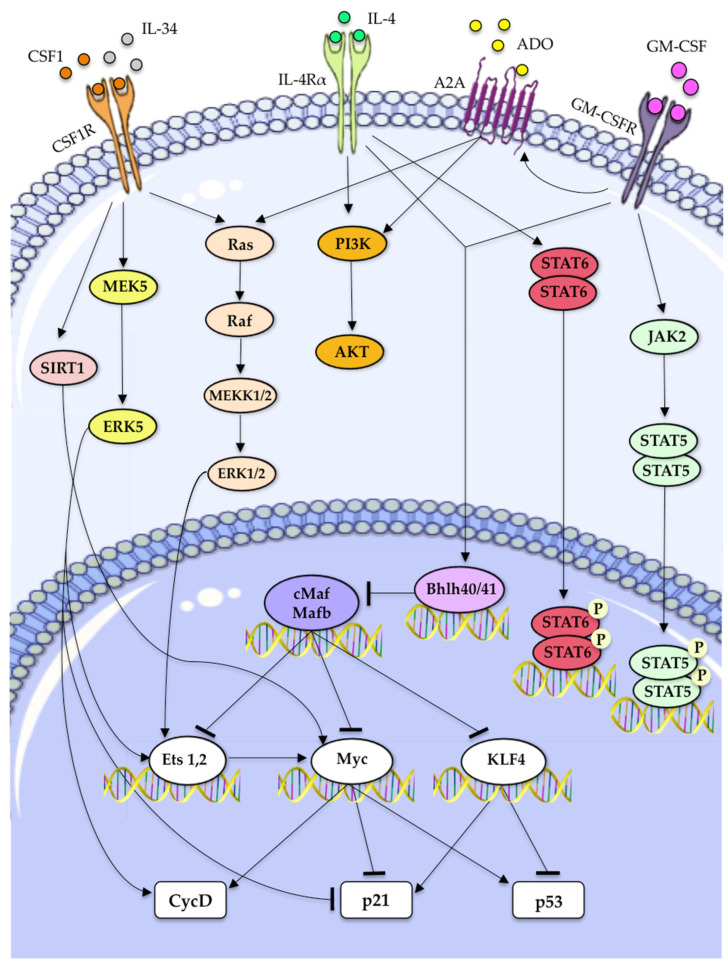
Signaling pathway involved in macrophage self-renewal. In response to CSF-1 and GM-CSF, SIRT1 regulates macrophage self-renewal through c-Myc and E2F-dependent pathways [125]. Soluble CSF-1R ligands (CSF1 and IL-34) trigger the Ras/MEK1/2/ERK1/2 pathway, which, in turn, activates Ets transcription factors increasing the expression of cyclin-D and c-Myc [12,125]. In addition, by recruiting Src family kinases (SFK), CSF-1R also induces the MEK5/ERK5 pathway leading to macrophage proliferation [126]. Indeed, MEK5/ERK5 activation support TAMs proliferation by suppressing p21 expression [63]. IL-4 can induce macrophage proliferation through STAT6-mediated transcription [43] and activates PI3K/AKT signaling, which is important for its pro-proliferative activity [12]. In response to GM-CSF, the JAK2/STAT5 pathway is triggered inducing macrophage proliferation [14,67]. Moreover, IL-4 and GM-CSF activate Ets, c-Myc, and KLF4 by inhibiting c-Maf and MafB expression downstream of the Bhlhe40/41 transcription factors [107,108] and work together with the ADO/A2A pathway to induce macrophage self-renewal via PI3K/Akt and MEK1/2/ERK1/2 signaling [106]. The molecular mechanisms of other signaling pathways, such as TREM2/DAP12, in macrophage proliferation remain to be documented. PI3K, phosphatidylinositol 3-kinase; A2A, ADORA2A adenosine receptor.

**Figure 4 biomedicines-10-02709-f004:**
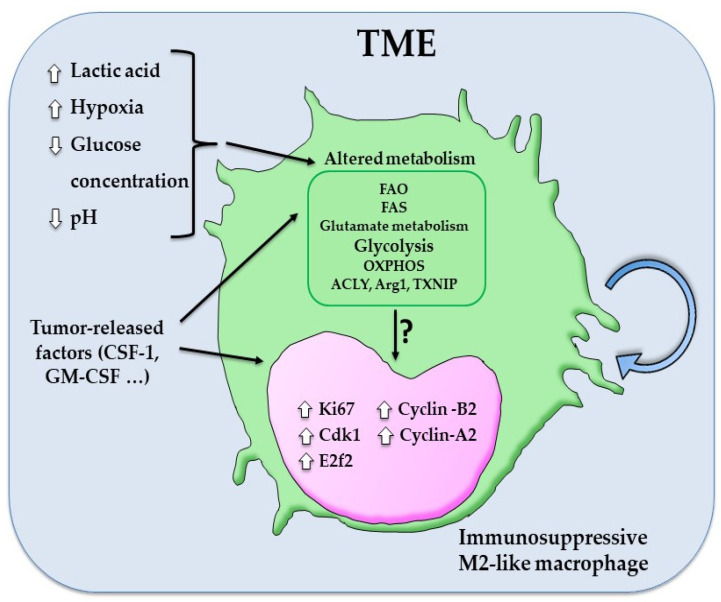
TME factors and metabolic signature are associated to TAMs self-renewal. Altered metabolic profile of TME, characterized by high concentrations of lactic acid, increased hypoxia, low glucose concentration, and reduction of pH, induces the immunosuppressive M2-like phenotype in macrophages [161,162]. TAMs show a unique metabolism capacity dependent to FAS, FAO, OXPHOS, altered glycolysis, and glutamate metabolism, as well as a high increase of metabolic enzymes such as Thioredoxin-interacting protein (TXNIP), ATP Citrate Lyase (ACLY), and Arg1 [160]. This specific metabolic phenotype of TAMs, which is induced by mitogenic stimuli, such as CSF-1 [145,166,167], is associated with an increase of several cell cycle regulators, including cyclin-dependent kinase Cdk1, cyclin-A2 and -B2, and E2f2 [168]. However, how the metabolic reprogramming directly regulates TAMs self-renewal remain to be elucidated.

**Table 1 biomedicines-10-02709-t001:** Molecular factors cancer-specifically driving self-renewal of macrophages.

Molecular Factor	Mechanism to Drive Macrophages Self-Renewal	References
CSF-1	• ERK1/2 activation, inducing the activation of Ets transcription factors, thus increasing of cyclin-D expression.	[12,78].
	• ERK5 activation, downregulation of p21 expression and increased expression of Ets2 and Cyclin-D	[63]
	• KLF2 and KLF4 upregulation.	[13]
	• c-myc upregulation.	[13]
IL-34	• Partially affects TAMs proliferation through its ability to interact with CSF1R.	[99]
GM-CSF	• Activation of JAK2/STAT5 pathway.	[12,13,14]
	• Activation of Bhlhe40/Bhlhe41 which inhibits c-Maf and Mafb transcription factors, thereby directly promoting expression of transcripts encoding cell cycle-related proteins.	[107,108]
ADO-1	• Working synergically with GM-CSF causes the upregulaion of PI3K/Akt and MEK/ERK signaling downstream of A2A activation.	[106]
	• Promotes macrophage proliferation, in part, through the transcriptional regulation of A2A.	[106]
IL-4	• Induce the transcription of STAT6 and activates PI3K/AKT signaling.	[12]
	• May stimulate macrophage proliferation in an Akt- and ACLY-dependent manner.	[109]

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
