# Peer review of "Self-Renewal of Macrophages: Tumor-Released Factors and Signaling Pathways"

_biomedicines, 2022, doi:10.3390/biomedicines10112709_

Round 1

Reviewer 1 Report

This review distills wide-ranging information about the origin and roles of tumor associated macrophages.  It makes it clear why there is no simple resolution about the effects in any given case.  At the same time it highlights the potential targets for therapy that may be exploited.   The review is simultaneously comprehensive and succinct.  It is well written and will be educational for readers at all levels of expertise in this area.

Author Response

We thank the Reviewer for her/his positive comments.   

Reviewer 2 Report

Filiberti and colleagues discuss the translational relevance of an important cellular phenomenon: macrophage renewal in their review, focusing specifically on the TME. The topic is intriguing and therefore, highly relevant. The paper is well-crafted, concise, and highly informative. I have the following suggestions to improve the manuscript before publication: 

- In the title the phrasing „Macrophages self-renewal”and in other titles „Macrophages xy… „ is grammatically not so correct. Please correct these instances to „Self-Renewal or xy of macrophages” or „Macrophage self-renewal or xy”.

- Every review should have a comprehensive, informative Table, also this one to summarize the most important molecular factors cancer-specifically driving macrophage self-renewal. 

- Summarizing the current preclinical and clinical endeavors aiming to modulate macrophage survival- and renewal should deserve a short section (max 300 words). The authors mention some clinically relevant targets in every section, however, systematically listing these studies separately should make a good addition to the manuscript. Authors should cite here studies rather in the clinical field (those of basic research have been cited exhaustively in previous sections)

Author Response

Filiberti and colleagues discuss the translational relevance of an important cellular phenomenon: macrophage renewal in their review, focusing specifically on the TME. The topic is intriguing and therefore, highly relevant. The paper is well-crafted, concise, and highly informative. I have the following suggestions to improve the manuscript before publication: 

- In the title the phrasing „Macrophages self-renewal”and in other titles „Macrophages xy… „ is grammatically not so correct. Please correct these instances to „Self-Renewal or xy of macrophages” or „Macrophage self-renewal or xy”.

Response. Thank you for the corrections. As requested, the titles have been corrected.  

- Every review should have a comprehensive, informative Table, also this one to summarize the most important molecular factors cancer-specifically driving macrophage self-renewal. 

Response. As suggested, the cancer-specific molecular factors driving macrophages self-renewal have been reported in table 1.

- Summarizing the current preclinical and clinical endeavors aiming to modulate macrophage survival- and renewal should deserve a short section (max 300 words). The authors mention some clinically relevant targets in every section, however, systematically listing these studies separately should make a good addition to the manuscript. Authors should cite here studies rather in the clinical field (those of basic research have been cited exhaustively in previous sections)

Response. We thank Reviewer 2 for this suggestion. Based on the recent literature, clinical trials targeting TAMs are based on strategies modulating monocytes/macrophage recruitment or depleting/reprogramming differentiated TAMs. To our knowledge, approaches blocking macrophage renewal are totally missing. We have now better specified and integrated this part of the Review in the conclusions section, as following: “During the past decade, the explosive growth in macrophage-targeted therapy indicate the TAM-targeted therapy is an effective antitumor strategy, especially as a complementary strategy in combination with conventional chemotherapy, radiotherapy, or immunotherapy. Most of the preclinical studies and clinical trials have been based on the therapeutic strategies according to their different mechanisms, including those that inhibit mononuclear macrophage recruitment or deplete/reprogram terminally differentiated TAMs (as reviewed in PMID: 35317518 and PMID: 36031601). However, clinical trials specifically designed to modulate or interfere with the self-renewal of TAMs are lacking”.